Removal of Microcystis aeruginosa cells using the dead cells of a marine filamentous bacterium, Aureispira sp. CCB-QB1

Furusawa Go furusawa@usm.my 1
Iwamoto Koji 2
1 Centre for Chemical Biology, Universiti Sains Malaysia , Bayan Lepas , Penang , Malaysia
2 Malaysia-Japan International Institute of Technology , Kuala Lumpur , Wilayah Persekutuan Kuala Lumpur , Malaysia
Anderson Todd
Electronic publication date: 2022 Feb 21
Publication date: 2022
Volume: 10
Electronic Location ID: e12867
Received 2021 Oct 22; Accepted 2022 Jan 10
Copyright: ©2022 Furusawa and Iwamoto
Copyright year: 2022
Copyright holder: Furusawa and Iwamoto
License: This is an open access article distributed under the terms of the Creative Commons Attribution License, which permits unrestricted use, distribution, reproduction and adaptation in any medium and for any purpose provided that it is properly attributed. For attribution, the original author(s), title, publication source (PeerJ) and either DOI or URL of the article must be cited.
License URL: https://creativecommons.org/licenses/by/4.0/

Keywords: Microcystis aeruginosa, Aureispira sp, Bioflocculant, CaCl2, FeCl3

Funding: Universiti Sains Malaysia (USM-Strategic Initiative-Ten Q1Q2, Project No. 311.PCCB.411954) This project was financially supported by Universiti Sains Malaysia (USM-Strategic Initiative-Ten Q1Q2, Project No. 311.PCCB.411954). The funders had no role in study design, data collection and analysis, decision to publish, or preparation of the manuscript.

==============================
Inorganic and synthetic flocculants are widely investigated for removing harmful microalgae, such as Microcystis aeruginosa. However, their toxicity and non-biodegradability are shortcomings. Bioflocculants based on extracellular polysaccharides have attracted much attention as alternative flocculants. However, its high production cost is a limiting factor for applying bioflocculants. Here, we investigate the potential of the dead cells of a marine filamentous bacterium, Aureispira sp. CCB-QB1, as a novel flocculant on M. aeruginosa cells. The removal efficiency of M. aeruginosa cells by the dead cells was measured by mixing and shaking both components in a buffer with 5 mM CaCl2 in different incubation times and concentrations of the dead cells. After that, the minimum effective concentration of CaCl2 was determined. The combination effect of FeCl3 and the dead cells on the removal efficiency was tested. The structure of cell aggregates consisted of the dead cells and M. aeruginosa cells were also observed using a scanning electron microscope. The maximum removal efficiency (75.39%) was reached within 3 min in the presence of CaCl2 when 5 mg/ml of the dead cells (wet cells) were added. The optimal concentration of CaCl2 was 5 mM. The combination of the dead cells and a low concentration of FeCl3 (10 mg/L) with 5 mM of CaCl2 significantly improved the removal efficiency by about 1.2 times (P < 0.05). This result indicates that the combination usage of the dead cells can reduce the use of FeCl3. These results indicated that the dead cells could potentially be a novel biolfocculant to remove M. aeruginosa cells.

Introduction

Harmful blooms of cyanobacteria (cyanoHABs) by toxic cyanobacterial species have been recognized in many countries (Briand et al., 2003) and generate direct economic damages because of the pollution of freshwater environments, as sources of drinking water, agriculture, fishing, and industry water (Hamilton, Salmaso & Paerl, 2016). CyanoHABs also seriously affect health problems in humans and animals because of the production of toxins such as microcystin and anatoxin (Azevedo et al., 2002; Katırcıoğlu, Akin & Atici, 2004). For instance, microcystin and its derivatives disrupt the structure and function of the liver and cause deaths by liver hemorrhage (Azevedo et al., 2002; Katırcıoğlu, Akin & Atici, 2004). CyanoHABs occur in many countries in tropical and temperate regions. In addition, it is known that climate change, such as rising CO2 and global warming, is likely to stimulate the development of harmful cyanobacterial blooms in eutrophic waters (Visser et al., 2016).

To remove microalgae, including cyanobacteria cells, from the water column, several types of flocculants are used. The flocculants are mainly classified into three groups, such as inorganic flocculants (e.g., polyaluminum chloride (PAC) and ferric chloride (FeCl3)), organic synthetic flocculants (e.g., polyacrylamide derivatives), and some kinds of clay or bioflocculants mainly consisting of polysaccharides produced by microorganisms (Salehizadeh & Shojaosadati, 2001). Inorganic flocculants, such as PAC and FeCl3, were low-cost materials and exhibited a high removal efficiency on M. aeruginosa cells (Li et al., 2015; Sun et al., 2012; Zhang et al., 2011). However, a high dosage of the flocculants is required for its high removal efficiency. In addition, the sludge containing inorganic flocculants consisted of heavy metals is toxic to the environment (Fast, Kokabian & Gude, 2014). Especially, aluminum may be one of the risk factors for Alzheimer’s disease (Flaten, 2001; Fast, Kokabian & Gude, 2014).

Polyacrylamide derivatives are extensively used to treat natural water and sewage because of their high removal efficiency on pollutants (Kurenkov, Hartan & Lobanov, 2002). Polyacrylamide derivatives are also used in removing microalgae, including M. aeruginosa (Pugazhendhi et al., 2019; Cai et al., 2021). However, acrylamide is considered a probable human carcinogen and mutagen in vitro and in vivo experiments using animals (Dearfield et al., 1988; Dearfield et al., 1995). Thus, there are concerns about their toxicity and less biodegradability.

To date, bioflocculants, which are non-toxic and biodegradable compounds, have attracted attention as an alternative to chemical flocculants. Bioflocculants are biological polymers, mainly extracellular polysaccharides (EPS) or proteins, produced by living organisms, particularly plants and microbes. As the plant-based natural coagulants, some seeds and fruit/ peels, such as Cicer arietinum, Moringa oleifera, and cactus, are reported (Kumar, Othman & Asharuddin, 2017). Bioflocculants composed of EPS produced by bacteria belonging to genera Bacillus (Deng et al., 2003; Zheng et al., 2008), Corynebacterium (He, Li & Chen, 2004), Klebsiella (Wang et al., 2007), Paenibacillus (Li et al., 2013), Proteus (Xia et al., 2008) and Pseudomonas (Azzam & Tawfik, 2015; Liu et al., 2016) were also found during past decades. These bioflocculants were exhibited high removal efficiency (80–99.6%) on negatively charged clay particles, kaolin. In addition, several investigations demonstrated that the bioflocculants were helpful for removing M. aeruginosa. For instance, Sun and colleagues reported that bioflocculant, EPS-1, produced by Bacillus amyloliquefaciens could remove approximately 88% of M. aeruginosa cells (Sun et al., 2015a). Bioflocculant produced by Pseudomonas aeruginosa ZJU1 combined with kaolin, CaCl2, and KAI(SO4)2 also exhibited high removal efficiency on M. aeruginosa cells (Sun et al., 2015b). The investigation conducted by Zhang et al. (2019) demonstrated that the bioflocculant produced by Halobacillus sp. strain H9 also successfully removed M. aeruginosa cells. Besides plants and microbes, another type of polysaccharide, chitosan, produced from shells of crustaceans has also attracted much attention as a bioflocculant to remove M. aeruginosa. The study conducted by Park et al. (2020) demonstrated that chitosan fiber removed almost 89% of M. aeruginosa cells. Ma et al. (2016) also reported that chitosan-aluminum chloride combined coagulants exhibited high removal efficiency (>90%) on M. aeruginosa cells. Thus, bioflocculants consisted of non-toxic and biodegradable biopolymers might be useful for removing M. aeruginosa cells. However, the high cost of the natural macromolecules, especially EPS from microorganisms, limits their application.

In a previous investigation, we reported that a marine filamentous bacterium, Aureispira sp. CCB-QB1, formed cell aggregates in the presence of CaCl2 (Furusawa, Hartzell & Navaratnam, 2015). In addition, the dead cells of Aureispira sp. CCB-QB1, which has a negatively charged cell surface, could absorb negatively charged clay particles, kaolin, and form aggregates with kaolin particles in the presence of CaCl2 (Hasyimah, Furusawa & Amirul, 2021). Our investigation demonstrated that the precipitation might be occurred by neutralizing the negative charge of both kaolin and the dead cells by adding a divalent cation, CaCl2. Based on the result, we predicted that M. aeruginosa cells might also be precipitated by the dead cells in the presence of CaCl2 (Hasyimah, Furusawa & Amirul, 2021). Furthermore, the preparation of the dead cells was more straightforward than that of known bioflocculants consisted of polysaccharides because the precipitation and purification steps required in known bioflocculants were not necessary to prepare the dead cells. This information indicates that the dead cells have the potential for an inexpensive flocculant compared with the other bioflocculants based on extracellular polymers. In this study, to confirm our prediction, we performed to remove M. aeruginosa cells by using the dead cells from aqueous samples under various conditions. To the best of our knowledge, this is the first report to remove the M.aeruginosa using the dead cells of a filamentous bacterium.

Material and Methods

Microorganisms and culture conditions

Aureispira sp. CCB-QB1 (referred to hereafter as CCB-QB1) deposited in CCB Microbial Biodiversity Library (CCB-MBL) (No. CCB-MBL0192) was the cells used as a bioflocculant in this study. The cells were cultured in a modified high nutrient artificial seawater medium (H-ASWM) (0.91% tryptones, 3.39% artificial sea salt, 10 mM 4-(2-hydroxyethyl)-1-piperazineethanesulfonic acid (HEPES), pH 7.6) described by Hasyimah, Furusawa & Amirul (2020) at 30 °C for 20 h with shaking (200 rpm). Microcystis aeruginosa NISE 102 strain (referred to hereafter as NISE 102) (Gao et al. (2012)) was obtained from the Microbial Culture Collection at the National Institute for Environmental Studies (NIES Collection, Tsukuba, JAPAN). The NISE 102 cells were cultured in liquid BG-11 medium at 26 °C and under a fluorescent lump at 3000 Lux, with a light: dark period of 12:12 h for 14 days.

Preparation of the dead cells of CCB-QB1

The dead cells of CCB-QB1 were prepared based on the method described by Hasyimah, Furusawa & Amirul (2020). To obtain exponential phase cells, the CCB-QB1 cells were cultured in 100 ml of modified H-ASWM at 30 °C for 20 h with shaking. An optical density at 600 nm (OD600) of cell suspension was measured using a spectrophotometer (UV-1800, Shimadzu, Japan) and adjusted OD600 = 1.0 with modified H-ASWM broth. The cell suspension was exposed to 3 W/m2 of UV-A in a laminar flow cabinet (AVC-4D, Esco Technologies Inc., USA) for 10 min, and then the suspension was centrifuged at 3,200 × g for 5 min to harvest cells.

Observation of aggregation formation of the dead cells on NISE 102 cells

NISE 102 cells in the exponential phase were centrifuged at 1,000 × g for 10 min. The cell pellet was suspended into 7 mM CaCl2 solution with 10 mM HEPES buffer (pH 7.6) and adjusted to OD680 = 0.6 (4.15 × 106 ± 5 × 104 cells/ml). The HEPES buffer was used as a basal buffer for preparing the cell suspension of NISE 102. The wavelength was referred from Sun’s report (Sun et al., 2015a; Sun et al., 2015b). Fifty mg of the dead cells (wet cells) was added to 10 ml of NISE 102 cell suspension. The suspensions were shaken for 3 min at 30  °C with shaking (100 rpm). The aggregates were observed under a light microscope (Olympus BX51, Olympus, Japan).

Effect of incubation time and initial concentration of the dead cells on the removal efficiency

One hundred mg, 50 mg, and 25 mg of the wet dead cells were added to 10 ml aliquots of NISE 102 cells suspension (OD680 = 0.6). The final concentration of the wet dead cells is 10 mg/ml, 5 mg/ml, 2.5 mg/ml and 1 mg/ml in 10 ml of NISE 102 cells suspension, respectively. The suspensions were shaken at 100 rpm for 3, 10, and 30 min at 30 °C in a shaker incubator. The aggregates were settled down at the bottom of the flask for 5 min. After that, 1 ml of the supernatant was carefully removed and transferred to a 1 ml cuvette. The absorbance was measured by a UV spectrophotometer at a wavelength of 680 nm. The removal efficiency was calculated as follows: Removal efficiency%=1−A/0.6×100.

Where A is OD680 of samples. 0.6 indicates the initial optical density of NISE 102 cells.

Effect of CaCl2 on the removal efficiency

Five mg/ml of the dead cells (wet cells) was added to NISE 102 cell suspensions (OD680 = 0.6) containing 50 mM, 20 mM, 5 mM, and 1 mM of CaCl2. The suspensions were shaken at 100 rpm for 3 min at 30 °C in a shaker incubator, and the aggregates were settled down at the bottom of the flask for 5 min. The removal efficiency was calculated by the same method described above.

Effect of FeCl3 on the removal efficiency

One hundred mg/L, 50 mg/L, 25 mg/L, and 10 mg/L of FeCl3 were prepared in NISE 102 cell suspensions adjusted at OD680 = 0.6 with/without 5 mM CaCl2. The aggregate formation and the removal efficiency were calculated using the procedure described above. Next, to measure the effectiveness of the combination of the dead cells and FeCl3 on the removal efficiency, 10 mg/L of FeCl3 with 5 mg/ml of the dead cells (wet cells) was prepared in NISE 102 cell suspensions (OD680 = 0.6) containing 5 mM CaCl2. The aggregate formation and calculation of the removal efficiency were conducted using the same procedure described above

Scanning electron microscope (SEM) observation of aggregates

Fifty mg of the dead cells were added into 10 ml of the suspension of NISE 102 cells (OD680 = 0.6) with CaCl2 and mixed at 100 rpm for 3 min. In a sample with FeCl3, 10 mg/L of FeCl3 was prepared in the suspension after mixing both cells for 3 min. The suspension was also mixed at 100 rpm for 3 min to make samples for SEM observation. The freeze-drying method was used for preparing the SEM samples. The aggregates were transferred to aluminum plates, and the extra water from the samples was carefully removed. The plates were placed on wet filter paper on a petri dish. For fixing the samples, a few drops of 2% Osmium tetroxide (OsO4) were dropped on the filter paper, and the petri dish was immediately closed and left in the fume hood for about 1 h. After that, the samples were frozen in liquid nitrogen for a few seconds and dried using a freeze-drying machine (Emitech K750X, Emitech, UK). Finally, the samples were coated with gold and observed by a Leo Supra 50 VP field emission scanning electron microscope (Carl Zeiss, Germany).

Statistical analysis

All results were represented by the mean ± standard deviation (SD). All data were subjected to one-way analysis of variance (ANOVA) with Tukey tests to examine significant differences between individual mean values. All statistical tests were carried out using SPSS Version 27.0 (IBM). Differences were considered statistically significant at P values <0.05. Independent t-test was used to compare the differences between samples with and without CaCl2 and samples with and without FeCl3. The analysis was also conducted by using SPSS Version 27.0.

Figure 1 Observation of cell aggregates consisted of the dead cells and NISE 102 cells in the presence of 7 mM CaCl2.

(A) Aggregation formation by the dead cells with NISE 102 cells. Top is the sample without CaCl2. Bottom is the sample with CaCl2. (B) Light microscopic observation of both the dead cells and NISE 102 cells (Top) and the inside of the aggregate (Bottom) (×1,000). Top is the sample without CaCl2. Bottom is the sample with CaCl2. The scale bar (A) one cm, (B) 20 µm.

Results

Aggregate formation of the dead cells with NISE 102 cells

To test the absorption ability of the dead cells of CCB-QB1 against NISE 102 cells, the dead cells of CCB-QB1 and NISE 102 cells were mixed in 10 mM HEPES buffer (pH 7.6) with and without 7 mM CaCl2. Figure 1A showed that both cells formed small aggregates within 3 min incubation period only in the presence of CaCl2. In contrast, the cell aggregates were not observed in the absence of CaCl2 (Fig. 1A). In light microscopic observation, isolated cells of NISE 102 and QB1 were observed in the absence of CaCl2 (Fig. 1B, top). On the other hand, NISE 102 cells were surrounded and trapped by many CCB-QB1 cells in the presence of CaCl2 (Fig. 1B, bottom), indicating that the dead cells of CCB-QB1 are capable of removing NISE 102 cells.

Figure 2 Effect of incubation time and dosage of the dead cells on removal efficiency of NISE 102 cells. 1.0, 2.5, 5.0, and 10.0 mg/ml of the dead cells (wet cell) with 7 mM CaCl2 were mixed with NISE 102 cells for 3, 10, and 30 min with shaking (100 rpm).

Different letter indicates a significant difference (One-way ANOVA and Tukey’s analysis, P < 0.05).

Effect of incubation time and dosage of the dead cells on removal efficiency

Generally, incubation time and initial concentration of flocculant are critical factors for forming flocs and removing the target. Hence, the removal rate of NISE 102 cells was measured using 10.0, 5.0, 2.5, and 1.0 mg/ml of the dead cells for each incubation period (3, 10, and 30 min). For 3 min incubation, the removal efficiency of 10.0 and 5.0 mg/ml of the dead cells was 76.77 and 75.39%, respectively (Fig. 2). The removal rate decreased with decreasing the dosage of the dead cells, such as 71.89 (2.5 mg/ml) and 68.06% (1.0 mg/ml) (Fig. 2). Especially, the removal efficiency of the sample with 1.0 mg/ml was significantly lower than that of the sample with 10.0 and 5.0 mg/ml (P < 0.05). As shown in Fig. 2, samples incubated for 10 and 30 min also exhibited similar tendencies. The samples added of 1.0 mg/ml of the dead cell exhibited the lowest removal efficiency (68.06 and 66.83%, respectively) in both samples. In addition, the removal efficiency of the samples with 10.0 and 5.0 mg/ml of the dead cells was similar to among the three different incubation periods. These results indicated that 3 min incubation period was enough to remove NISE 102 cells from the aqueous sample. In addition, the removal efficiency between 10.0 and 5.0 mg/m samples after 3 min incubation period was not significantly different (P > 0.05). Therefore, the following experiments were conducted with 5.0 mg/ml of the dead cells for 3 min incubation period.

Effect of CaCl2 on the removal efficiency

To confirm the effect of CaCl2 against the removal of NISE 102 cells, 50, 20, 5, and 1 mM of CaCl2 were added to the suspension of NISE 102 cells. Although we expected that high CaCl2 concentration conditions, 50 mM and 20 mM, stimulated robust removal of the cells compared to 5 mM samples, the removal rate of the three samples was quite similar (75. 11, 76.44, and 75.44%, respectively) (P > 0.05) (Fig. 3). On the other hand, the sample added of 1 mM of CaCl2 exhibited significantly lower removal efficiency (30.05%) (P < 0.05) than those of samples containing a higher concentration of CaCl2 (Fig. 3). The results showed that the minimum effective concentration of CaCl2 for removing NISE 102 cells was 5 mM, and the concentration was used in the following experiments.

Figure 3 Effect of calcium ion on the removal efficiency of NISE 102 cells.

1, 5, 20, 50 mM of CaCl2 were added into NISE 102 cell suspension with 5.0 mg/ml of the dead cell. The suspension was mixed for  3 min. Different letter indicates a significant difference (One-way ANOVA and Tukey’s analysis, P < 0.05).

Effect of CaCl2 and FeCl3 on the removal efficiency

First, four different concentrations of FeCl3, 100, 50, 25, and 10 mg/L, were added to the suspension of NISE 102 cells without dead cells of QB1. The removal efficiency of samples with 100 and 50 mg/L of FeCl3 exhibited 100 and around 90% in the absence and presence of CaCl2, respectively (Fig. 4A). On the other hand, as shown in Fig. 4A, the removal efficiency decreased with decreasing the concentration of FeCl3. The samples with 25 and 10 mg/L of FeCl3 removed 61.67 and 40.23% of the cells without CaCl2 (Fig. 4A). Interestingly, the removal efficiency increased with adding the 5 mM CaCl2 in the samples with 25 and 10 mg/L of FeCl3 as 76.44 and 57.61%, respectively (Fig. 4A). Especially, the removal rate with 10 mg/L of FeCl3 with CaCl2 was 1.43 times higher than that without CaCl2 (P < 0.05). This result showed that the removal efficiency with 10 mg/L of FeCl3 was improved by adding 5 mM CaCl2.

In Fig. 4B, the removal efficiency of 10 mg/L FeCl3 with the dead cell on NISE 102 cells was tested. The removal efficiency of FeCl3 with the dead cell was 70.28%. Expectedly, the removal efficiency was significantly improved from 70.28% to 84.17% by adding 5 mM CaCl2 (P < 0.05) (Fig. 4B).

SEM observation of cell aggregates

The aggregates were observed under SEM to investigate how dead cells formed the aggregates with NISW 102 cells. In the sample without FeCl3, big aggregates were observed (Fig. 5A), and many NISE 102 cells were trapped in the aggregates (Fig. 5B). The dead cells that possessed filamentous structures got entangled in each filamentous cell. The dead cells tightly attached NISE 102 cells, and the entangled cells captured NISE 102 cells (Figs. 5C and 5F). Big aggregates were also observed in the sample with FeCl3 (Fig. 5D). Many NISE 102 cells were also trapped in the aggregates as well as the sample without FeCl3 (Fig. 5E). Compared to the sample without FeCl3, the cells were packed more tightly in the aggregates (Fig. 5F).

Figure 4 Effect of CaCl2 and FeCl3 on the removal efficiency of NISE 102 cells.

(A) The removal efficiency of NISE 102 cells on different concentration of FeCl3 (10, 25, 50, 100 mg/ml) without the dead cells. Blue bars indicate samples without CaCl2, Orange bars indicate samples with CaCl2. (B) The removal efficiency of NISE 102 cells with the dead cells in the presence of FeCl3 (10 mg/ml) with and without 5 mM CaCl2. Asterisks in (A) and (B) indicate the significant difference (Independent t-test, P < 0.05).

Discussion

In this study, we tried to apply the dead cells of CCB-QB1 for removing harmful algae, M. aeruginosa. Our previous investigations demonstrated that CCB-QB1 cells aggregated in the presence of 7 mM CaCl2 (Furusawa, Hartzell & Navaratnam, 2015), and the dead cells of CCB-QB1 formed aggregates with kaolin clay particles in the same condition (Hasyimah, Furusawa & Amirul, 2021). In keeping with these data, we expected that the dead cells of CCB-QB1 could also form aggregates and precipitate on M. aeruginosa in the presence of CaCl2. To confirm the hypothesis, the dead cells of CCB-QB1 and NISE 102 cells were mixed in the buffer with or without CaCl2. As shown in Fig. 1A, although no cell aggregates were observed in the absence of CaCl2, the dead cells could make aggregates with NISE 102 cells in the presence of CaCl2. In addition, the removal efficiency of NISE 102 cells decreased under low CaCl2 concentration (1 mM) (Fig. 3), indicating that CaCl2 is crucial for removing NISE 102 cells. Several investigators described that the cell surface of M. aeruginosa is negatively charged in water (Hadjoudja, Deluchat & Baudu, 2010; Shi et al., 2016). Generally, the net charge of the bacterial cell surface is also negative (Corpe, 1970; Li et al., 2019; Pajerski et al., 2019). Our previous investigations also indicated that the cell surface of CCB-QB was negatively charged (Furusawa, Hartzell & Navaratnam, 2015). However, the dead cells of CCB-QB1 could adsorb negatively charged materials, such as kaolin clay particles, in the presence of CaCl2 because Ca2+ might help neutralize the electrical forces (Hasyimah, Furusawa & Amirul, 2021). Sato and colleagues described that the addition of EPS produced by cyanobacteria and 1,000 mg/L of calcium (approximately 9 mM) was stimulated the colonization of M. aeruginosa cells (Sato et al., 2017). The addition of EPS and calcium decreased the zeta potential of M. aeruginosa cells and promoted its colonization. However, when only EPS was added, the zeta potential hardly decreased (Sato et al., 2017), resulting in inhibiting the colonization. These investigations revealed that Ca2+ plays an important role in neutralizing the charge of M. aeruginosa cells. Thus, it is expected that the negative charge of NISE 102 and the dead cells might also be neutralized by Ca2+, and the neutralization may eliminate repulsive electron forces between the dead cells and NISE 102 cells. As a result, Ca2+ might facilitate the biosorption of the dead cells against NISE 102 cells.

FeCl3 was used as a flocculant for removing M. aeruginosa cells from aqueous samples (Li et al., 2015; Hao et al., 2016; Gao et al., 2012). In this study, we also performed to use FeCl3 as a coagulant. Figure 4A showed that the removal efficiency of NISE 102 cells by a low concentration of FeCl3 was enhanced by adding CaCl2. Figure 5F also showed that the dead cells were more tightly aggregated in the presence of CaCl2 and FeCl3. The absorption of Fe(OH)3 generated from FeCl3 by organic matters could explain this result (Amuda & Amoo, 2007). Besides that, Liu et al. reported that Ca2+ increased the zeta potential of Fe(OH)3 and facilitated Fe(OH)3 floc aggregation (Liu et al., 2007). These investigations indicated that the increase of zeta potential and the promotion of floc aggregation by Fe(OH)3 via the addition of Ca2+ enhance the removal of NISE 102 cells. Figure 4A demonstrated that a high removal efficiency (>90%) of NISE 102 cells was observed when the dosage of FeCl3 was more than 50 mg/L. (Li et al., 2015) also described that the addition of 50 mg/L of FeCl3 exhibited almost the maximum removal efficiency on M. aeruginosa cells. Hao and colleagues also showed that M. aeruginosa cells could aggregate when the dosage of FeCl3 was 175 mg/L (Hao et al., 2016). Thus, the addition of a high dosage of FeCl3 was required to remove M. aeruginosa cells. Figure 4B demonstrated that the addition of only 10 mg/l of FeCl3 significantly improved the removal efficiency. These results also revealed that the combination of the dead cells and CaCl2 could reduce the dosage of FeCl3 for removing M. aeruginosa cells.

Sun and colleagues reported that maximum removal efficiency by bioflocculants from B. amyloliquefaciens strain DT and P. aeruginosa strain ZJU1 was reached within 10 min (Sun et al., 2015a; Sun et al., 2015b). The bioflocculant from Klebsiella pneumonia took 1.6 h for floc formation with M. aeruginosa cells (Xu et al., 2016). In contrast, in this study, the maximum removal efficiency by the dead cells was reached within 3 min (Fig. 2). This result revealed that the dead cells were able to quickly remove M. aeruginosa cells compared to other bioflocculants based on EPS from microorganisms. Another advantage of the usage of the dead cells is that the preparation process of the dead cells is more straightforward than that of other bioflocculants based on EPS from microorganisms. Generally, bioflocculant-producing bacteria were cultured for 2–3 days in a nutrient-rich broth. For instance, several types of saccharides (5.0 g/L sucrose, 2.0 g/L glucose, and 2.0 g/L maltose) were added to the culture broth to produce a bioflocculant from P. aeruginosa ZJU1 (Sun et al., 2015b). In addition, two volumes of cold ethanol were required to recover the bioflocculant released into the culture supernatant. Finally, the crude bioflocculant was centrifuged to harvest and dried by heat treatment or freeze-drying. In contrast, the CCB-QB1 cells were cultured in a seawater medium without adding any carbon sources. The incubation period to obtain the cells was 24 h (Hasyimah, Furusawa & Amirul, 2021). After 10 min UV treatment, the dead cells were able to use as a bioflocculant (Hasyimah, Furusawa & Amirul, 2021). Thus, the preparation procedure of the dead cells was quite simple in comparison with the other bioflocculants, suggesting that the production cost of the dead cells might be cheaper than that of known bioflocculants consisted of EPS from microorganisms.

Figure 5 Scanning electron micrographs of the aggregates consisted of the dead cells and NISE 102 cells.

(A, B, C): Samples without FeCl3. (D, E, F): Samples with FeCl3. The scale bars A: 100 µm (100×); B: 20 µm, (400×); C: 2 µm (1,500×); D, E: E; 2 µm (4,000×), 100 µm (100×); F: 10 µm (500×); G: 2 µm (1,500×).

It was reported that some plant-based natural coagulants also successfully removed M. aeruginosa cells. For example, protein-based coagulant extracted using a salt solution (1 M NaCl or 1 M CaCl2) from Moringa oleifera seeds exhibited a high removal rate (78.9%) on M. aeruginosa cells (Carvalho et al., 2016). Moreti and colleagues also reported that a saline extract from M. oleifera seeds could remove M. aeruginosa cells effectively (de Oliveira Ruiz Moreti et al., 2019). Interestingly, the extract could adsorb and remove microcystin-LR (de Oliveira Ruiz Moreti et al., 2019). The preparation procedure of the extract using saline solution was more straightforward than that of bioflocculants consisted of EPS. However, many pretreatment processes, such as dehusking, powdering, and drying, were required. Of course, the large agricultural land and long cultivating period to obtain seeds are necessary for culturing M. oleifera. In contrast, the biomass production of CCB-QB1 is faster and easier than the plant.

Conclusion

In this study, the dead cells of a marine filamentous bacteria, Aureispira sp. CCB-QB1, was used as a bioflocculant for removing a harmful freshwater microalga, M. aeruginosa. The dead cells were removed about 75% of M. aeruginosa cells with 3 min incubation in the presence of CaCl2. The optimization of CaCl2 concentration for removing the M. aeruginosa cells showed that the optimal concentration of CaCl2 was 5 mM. The removal efficiency by inorganic flocculant, FeCl3, was improved by the addition of CaCl2. Besides that, the combination of the dead cells and CaCl2 could reduce the dosage of FeCl3 for removing M. aeruginosa cells. These results indicated that the dead cells have the potential for a novel and alternative bioflocculant.

Supplemental Information

Supplemental Information 1 Effect of incubation time and dosage of the dead cells on removal efficiency of NISE 102 cells

1.0, 2.5, 5.0, and 10.0 mg/ml of the dead cells (wet cell) were mixed with NISE 102 cells for 3, 10, and 30 min with shaking (100 rpm). All data shown are mean values from three replicate experiments. Error bars denote the standard deviation of triplicate samples.

Click here for additional data file.

Supplemental Information 2 Effect of calcium ion on the removal efficiency of NISE 102 cells

1, 5, 20, 50 mM of CaCl2 were added into NISE 102 cell suspension with 5.0 mg/ml of the dead cell. All data shown are mean values from three replicate experiments. Error bars denote the standard deviation of triplicate samples.

Click here for additional data file.

Supplemental Information 3 Effect of CaCl2 and FeCl3 on the removal efficiency of NISE 102 cells

(a) The removal efficiency of NISE 102 cells on different concentration of FeCl3 (10, 25, 50, 100 mg/ml). (b) The removal efficiency of NISE 102 cells with the dead cells in the presence of FeCl3 (10 mg/ml). All data shown are mean values from three replicate experiments. Error bars denote the standard deviation of triplicate samples.

Click here for additional data file.

We are thankful to the Microscopy Unit in Universiti Sains Malaysia for SEM observation.

Additional Information and Declarations

Competing Interests

Author Contributions

Data Availability

The authors declare there are no competing interests.

Go Furusawa conceived and designed the experiments, performed the experiments, analyzed the data, prepared figures and/or tables, authored or reviewed drafts of the paper, and approved the final draft.

Koji Iwamoto conceived and designed the experiments, authored or reviewed drafts of the paper, and approved the final draft.

The following information was supplied regarding data availability:

The raw measurements are available in the Supplementary Files.

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
