# Peer review of "Removal of Microcystis aeruginosa cells using the dead cells of a marine filamentous bacterium, Aureispira sp. CCB-QB1"

_PeerJ, doi:10.7717/peerj.12867_

## Round 0.1 · original submission · Major Revisions

There were concerns about the experimental design, the lack of a detailed description of important parameters, and an apparent omission of both positive and negative controls. Further, there is no mention of the impact on water quality.

Additional comments/concerns from the reviewers (not repeated here) need to be addressed in your revision.

Finally, all 3 reviewers pointed out many grammatical errors in the manuscript. Please read the revised version carefully for grammar prior to re-submitting your revision.

·

Basic reporting

Long and listing up sentences are sometimes not so fluent (red tread) from sentence to sentence. Example:
As the plant-based natural coagulants, some seeds and fruit/ peels are reported such
90 as Cicer aretinum, Moringa oleifera, cactus, etc. (Kumar, Othman & Asharuddin, 2017). Several
91 bioflocculant-producing bacteria belonging to genera Bacillus (Deng et al., 2003; Zheng et al.,
92 2008), Corynebacterium (He, Li & Chen, 2004), Klebsiella (Wang et al., 2007), Paenibacillus (Li
93 et al., 2013), Proteus (Xia et al., 2008) and Pseudomonas (Azzam & Tawfik, 2015; Liu et al.,
94 2016) were found during past decades. These bioflocculants were exhibited high removal
95 efficiency (80 - 99.6 %) on negatively charged clay particles, kaolin. Several investigations
96 demonstrated that bioflocculants were useful for removing M. aeruginosa.
Check grammar, example:
100 … The investigation conducted by
101 Zhang et al. demonstrated that the bioflocculant produced by Halobacillus sp. strain H9 was also
102 successfully removed M. aeruginosa cells (Zhang et al., 2019).
This should be moved to discussion section:
Our previous investigations demonstrated that CCB-QB1 cells were aggregate in the
210 presence of 7 mM CaCl2 (Furusawa, Hartzell & Navaratnam, 2015), and the dead cells of CCB-
211 QB1 formed aggregates with kaolin clay particles in the same condition (Hasyimah, Furusawa &
212 Amirul, 2020). In keeping with these data, we expected that the dead cells of CCB-QB1 could also
213 form aggregates and precipitate
This should be moved to discussion or method section:
FeCl3 was used as a flocculant for removing M. aeruginosa cells from aqueous samples (Li
250 et al., 2015; Hao et al., 2016)(GAO, SHIMIZU & XUE, 2012). In this study, we also performed
251 to use FeCl3 as a coagulant.
Please avoid leading regarding result, this is an open question still as the result has not yet been shown and the method should be neutral
176 First, to confirm the removal
Avoid expectations
213 form aggregates and precipitate with M. aeruginosa cells. To confirm the expectation, the dead
Result, discussion and conclusion:
278 the presence of 7 mM CaCl2. In addition, the removal efficiency of NISE 102 cells decreased under
279 low CaCl2 concentration (1 mM) (Figure 3), indicating that CaCl2 is crucial for removing NISE
280 102 cells.
Why is the removal efficiency lower in figure 3: 5 mg/ml than all the samples in the figure without CaCl2 (Fig 1) 5 mM should be comparable? Isn’t figure 1 without CaCl2? Then there is a higher removal efficiency without CaCl2, or? So why does removal efficiency decrease with addition of CaCl2. Same effect is seen in 10 mg/L in figure 4a. Please provide a negative control in all figures.

Experimental design

Indicate replicates in the figure text also provide controls as described
Related to result, discussion and conclusion:
Aggregate formation of the dead cells with NISE 102 cells: please describe figure 1 in more detail
Figure 1:
It is not so clear that this is a binding of the two cells from this image. Provide positive and negative control
Also show that this could be repeated, indicate how may replicates were used
Figure 2:
3 min incubation. What is the indication that 1 mg/ml and 2.5 mg/ml is not significantly different, and therefor the choice of 5 mg/l to continue?
Figure 3
Provide negative control
Figure 4
Provide negative control. Better describe the difference of a and b in the figure text. Especially b, what is the condition compared to a, is there anything else in there and how is b related to a. Not clear in the figure text. Is 4a without dead cells? If so be more clear reg. difference.
Figure 5
Provide positive and negative control
Figure text,: what is d? “(a), (b), (c), (d) Samples without FeCl3. (d), (e), and (f) Samples with FeCl3”. It makes it not possible to know what the figure tells.

Validity of the findings

Please see points in section 2 for next check the validity of the findings. Elaborated further;
As mentioned in the introduction it should be tested also absorption of e.g heavy metals with this method
“116 also absorbed heavy metals, such as Fe3+ and Cu2+, and removed it from the solutions (Hasyimah,
117 Furusawa & Amirul, 2020).”
The efficiency vs incubation time measured after finding the correct combination of cells and chemical. This seem relevant to avoid adding high concentrations of CaCl2 and FeCl3. e.g. concentration vs time. Adding longer time might be more economically beneficial v.s adding chemicals.

Additional comments

The results seem promising for a method of increased sustainability in removal of M. aeruginosa cells that might be also extended in further studies.
However there are some essential parts, as mentioned that should be changed.
Especially in the results, see points and that should be provided e.g in order to analyze and validate the results.
Figures/result/discussion conclusion: There are no positive or negative controls. Please provide these data and other points made here, in order to evaluate the data, and change the description of result, discussion and conclusion accordingly.
Check language, flow between sentences, and some grammar (see examples in section 1),
use of italic
Abbreviations can be useful but sometimes confusing if used in other known concepts such as PAM.

Reviewer 2 ·

Basic reporting

1. The content and data of the article are insufficient and lack of scientific value. This work just evaluates the flocculation effect of M. aeruginosa by dead cells of Aureispira sp., did not explain why it can be flocculated, and what are the influencing factors of flocculation effect? In addition, the application area of this technology is water column. What is its impact on water quality? Do these dead cells have inhibitory effect on flocculated cyanobacteria?
2. There are too many clerical errors in the text. For example, (Hasyimah, Furusawa & A) in line 132; “5 gm/ml” in line 180 and 186; “5.0 mg/m” in line 233; “(Li et al., 2015; Hao et al., 2016)(GAO, SHIMIZU & XUE, 2012)” in line 250; “of FeCl3as” in line 257; “(Corpe, 1970; (Li et al., 2019; Pajerski et al., 2019)” in line 286, “flocculant, FeCl3, was improved by addition of CaCl2” in line 351. Please check the manuscript carefully before submitting it

Experimental design

The experimental design is puzzling. Only 10ml of algal liquid is used for flocculation test. What is the liquid depth in the flask, and can the supernatant be effectively separated? Please provide some photos of flocculation test. In addition, as the first experiment, how to determine the concentration of CaCl2 and dead cells in the section "observation of aggregation formation of the dead cells on NISE 102 cells". Please describe the purpose of the test in detail.

Validity of the findings

3. Two data graphs seem to be contradictory. Is the experiment in Figure 2 without CaCl2? The 5 mg/ml dead cells can remove more than 70% of cyanobacteria. However, the removal rate of cyanobacteria by 1 mM CaCl2 and 5 mg/mL dead cells in Figure 3 is only about 30%, which is puzzling.

Additional comments

Abstract:
1. The abstract is a summary of this study. Generally, "our previous report demonstrated" cannot appear in abstract. Also, in the conclusion
2. The abstract needs to be simplified. Generally, it is not necessary to list the “background”, “methods” and “results”, and the methods should be as brief as possible.
Methods and materials
1. It is not rigorous to use the mass of wet cells weighed after centrifugation as the addition amount, which will have a great error.
2. Line 142. Is H-ASWM the “high nutrient artificial seawater medium” mentioned above. Abbreviation should be marked when it first appears
3. Line 152. “5 mg/ml of the dead cells” should be replaced by “50 mg of the dead cells”.
4. Line 157-158. Is there BG11 medium or CaCl2 solution in the NISE 102 cell suspensions? In other words, what else is in the cell suspensions besides NISE 102 cells? This needs to be explained clearly, as well as in the following experiments
Discussion:
The discussion part needs to pay more attention on why dead cells can be used as flocculants to remove cyanobacteria, instead of why the addition of CaCl2 and FeCl3 can improve the removal efficiency.
Figures:
1. A control group without CaCl2 should be added in Figure 3.
2. The title of figure 4a should indicate whether there were dead cells, and what was the concentration of CaCl2. The title of figure 4b should indicate whether there was CaCl2 and what the concentration was.
3. “(a), (b), (c), (d) Samples without FeCl3” in the title of Figure 5 should be replaced by “(a), (b), (c) Samples without FeCl3”. In addition, no figure 5g.
Reference:
The order of references is not correct. If you want to follow the order in the text, you should replace “(… et al, ….)” with numbers, otherwise, you should order it alphabetically.

Reviewer 3 ·

Basic reporting

1. Language of the manuscript should be improved. Poor sentence formation with lot of grammatical mistakes are present in the entire manuscript.

2. Punctuation errors are present at various places in the manuscript, for e.g line 149, 151-152, 189, 192

3. Line 142- explain what is “modified H-ASWM”

4. Avoid starting a sentence with a number, e.g line 152,157, 186. Either frame your sentence in such a way so that it does not starts with a number or write the number in words.

5. Formatting errors are present, e.g line 282

6. All the references should be in uniform pattern. There are variations at various places.

Experimental design

Data indicating cell surface chemistry should also be included

Validity of the findings

No comment

---

## Round 0.2 · accepted · Accept

Thank you for responding to reviewer comments and revising your manuscript accordingly.

·

Basic reporting

The questions were answered according to the last review.

Experimental design

The questions were answered according to my last review and what was possible and relevant.

Validity of the findings

The questions were answered according to my last review and what was possible and relevant, I have no further comments.

Additional comments

Thanks for carefully going trough the comments, I have no further questions.